# Preoperative Neutrophil-Lymphocyte Ratio for predicting surgery-related acute kidney injury in non-cardiac surgery patients under general anaesthesia: A retrospective cohort study

Yongzhong Tang[1], Linchong Chen[1], Bo Li[2], Lin Yang[3], Wen Ouyang[1], Dan Li[1]*

1 Department of Anesthesiology, The Third Xiangya Hospital, Central South University, Changsha, China, 2 Operation Room, The Third Xiangya Hospital, Central South University, Changsha, China, 3 Department of Anesthesiology, The Second Xiangya Hospital, Central South University, Changsha, China

* danfuer623@163.com

**Data Availability Statement:** "Yes - all data are available without restriction" "The relevant data are

## Abstract

### Background

This study was conducted to investigate the relationship between the Neutrophil-Lymphocyte Ratio (NLR) and the incidence of surgery-related acute kidney injury (AKI) in non-cardiac surgery patients under general anaesthesia.

### Methods

In this retrospective study, 5057 patients from Third Xiangya hospital from January 2012 to December 2016 and 1686 patients from Second Xiangya hospital from January 2016 to December 2016 for non-cardiac surgery under general anesthesia were included. According to receiver operating characteristic (ROC) curve constructed by NLR for postoperative AKI, the cut-off point was obtained as the basis for grouping low or high NLR. The baseline characteristics of two sets were compared with each other. A multi-factor model was constructed by Least absolute shrinkage and selection operator (LASSO) method with the training set, and verified by outside validation set.

### Results

243 patients (3.604%) developed postoperative AKI. The ROC curve showed that the AUC of the NLR for predicting postoperative AKI in non-cardiac surgery was 0.743 (95% CI, 0.717–0.769), and the cut-off value was 3.555 (sensitivity, 86.4%; specificity 51.9%). There was no significant difference in the baseline characteristics of training set and validation set. The AUC in the training set was 0.817 (95% CI, 0.784–0.850), and the AUC in the validation set was 0.804 (95% CI, 0.749–0.858), the AUC deviation was 0.012 (P > 0.05) from validation set, and the likelihood ratio test showed P < 0.05.

available from Dataverse at doi:10.7910/DVN/
AEHZQD (https://doi.org/10.7910/DVN/AEHZQD)."

**Funding:** This work was supported by the National
Natural Science Foundation of China (grant number
82002097), Project of Health and Health
Commission of Hunan Province (grant number
20201802), Hunan Province Key Laboratory
Project (grant number 2018TP1009), and the
Natural Science Foundation of Hunan Province
(grant number 2020JJ5854. The funders had no
role in study design, data collection and analysis,
decision to publish, or preparation of the
manuscript.

**Competing interests:** The authors have declared
that no competing interests exist.

**Abbreviations:** NLR, Neutrophil-Lymphocyte Ratio;
AKI, acute kidney injury.

## Conclusion

This study showed that preoperative high NLR (NLR≥3.555) was an independent risk factor
associated with postoperative AKI (OR, 2.410; 95% CI, 1.371–4.335) in patients for non-car-
diac surgery under general anesthesia.

## Introduction

Acute renal injury (AKI) is a serious postoperative complication with high incidence rate. The
incidence of postoperative AKI has been found to range from 3–43% [1–4], inspite that mea-
sures had been taken to protect renal function postoperatively. AKI is also a predictor of long-
term prognosis, including chronic kidney disease, end-stage renal disease (ESRD), and cardio-
vascular disease (heart failure, myocardial infarction), as well as mortality [5, 6]. Currently,
markers widely used to evaluate acute renal injury (such as BUN, SCr, and urine volume, etc.)
are not sensitive enough to detect early mild impairment of renal function [7]. Other indica-
tors such as Cystatin (CysC), urinary microalbumin (mALB), and 1-microglobulin (1-mg),
although more sensitive, are also indicators after renal injury and expensive to carry out in
clinical practice. Therefore, the purpose of this study is to explore a cheap, simple and predic-
tive index of postoperative acute renal injury, so as to achieve early prevention, early treatment,
and protect the renal function to the maximum extent.

Inflammation is one of the major pathogenic mechanism for AKI [8]. The proportion of
neutrophils to lymphocytes, i.e., the neutrophil-lymphocyte ratio (NLR), before surgery is an
effective index reflecting inflammation and oxidative stress. A large number of studies have
shown that the NLR is closely related to the prognosis of patients with tumors [9, 10], and car-
diovascular disease [11–13] At present, studies have been conducted on NLR and specific
high-risk populations of AKI, such as emergency surgery [14], burn surgery [15], cardiac sur-
gery [16], and sepsis patients [17]. However, the relationship between the preoperative NLR
and AKI after non-cardiac surgery under general anaesthesia has been barely studied. This ret-
rospective study aimed to assess the significance of the preoperative NLR in predicting acute
renal injury in patients for non-cardiac surgery under general anaesthesia, and to obtain a cut-
off value of preoperative NLR to distinguish high-risk population and improve the sensitivity
of diagnosis.

## Materials and methods

### Patients

This retrospective study was performed at the Third Xiangya Hospital of Central South Uni-
versity from January 2012 to December 2016 (n = 5057) and the Second Xiangya Hospital of
Central South university from January 2016 to December 2016 (n = 1686). The inclusion crite-
ria were patients aged≥18 years who underwent non-cardiac surgery under general anesthesia.
Patients with a preoperative infection or chronic kidney disease were excluded, along with
those admitted for urinary surgery and those with missing data.

This study was in line with the guidelines of the Strengthening of Observational Epidemio-
logical Studies (STROBE) statement, and approved by the ethics committee of the Third Xian-
gya Hospital of Central South University (2017-S214). Because of observational nature of the
study, informed consent was waived by the Third Xiangya Hospital of Central South
University.

## Data collection

The following information was collected: 1. Epidemiological data including age and gender; 2. Preoperative laboratory data including absolute value or percentage of neutrophil and lymphocyte, haemoglobin, platelet distribution width, uric acid, albumin, total bilirubin, creatinine and eGFR calculated using the Chronic Kidney Disease epidemiology collaboration (CKD-EPI) formula [18]. 3. Comorbidities of all the patients (diabetes, coronary heart disease, peripheral vascular disease and hypertension). 4. Intraoperative data including the operation type (emergency, or laparoscopic), American Society of Anesthesiologists (ASA) grade, operative time, total amount of fluids infusion and blood loss during operation. 5. Postoperative outcomes such as admission to ICU and estimated 10-year survival rate (calculated with Charlson comorbidity index).

## Definitions

Postoperative AKI was defined according to the Kidney Disease: Improving Global Outcomes (KDIGO) 2012 creatinine criteria [19], as one of the following: an increase in serum creatinine by $\geq$0.3 mg/dL within 48h or a $\geq$1.5-times increase in serum creatinine from baseline within 7 postoperative days. The baseline serum creatinine level was calculated using the lowest level at preoperative day 7. The NLR was calculated by dividing the absolute neutrophil count by the absolute lymphocyte count. Surgical grade was classified using the surgical classification catalog constituted by the Chinese Ministry of Health, published in 2018.

## Statistical analysis

IBM SPSS (version 22.0) and R (version 2.12.0) software were used for the statistical analysis. The data are presented as the Median (M) and interquartile range (IQR). Wilcoxon rank sum test was used to compare non-normal distribution continuous variables, and the chi-square test was used to compare the composition ratio of classified data. All p-values < 0.05 were considered significant.

First, the ROC curve of preoperative NLR was built by R software and the best cut-off value was calculated. Taking the data from The Third Xiangya Hospital as the training set, the variables P < 0.1 and some clinical meaningful variables were included in the multi-variable model, and the simplified model was obtained by using the LASSO regression. The data from The Second Xiangya Hospital were taken as the validation set. The ROC curve was drawn to verify that the high preoperative NLR was an associated independent risk factor. Finally, likelihood ratio test was conducted, and the importance of high preoperative NLR was verified again

## Results

In total, 6743 patients were included, of whom 243 (3.604%) developed postoperative AKI. The ROC curve showed that the AUC of the NLR for predicting postoperative AKI in non-cardiac surgery was 0.743 (95% CI, 0.717–0.769; p < 0.01), and the cut-off value was 3.555 (sensitivity 86.4%; specificity 51.9%). It was suggested that preoperative NLR$\geq$3.555 might be associated with the increasing risk of postoperative AKI (Fig 1).

5057 patients from Xiangya Third Hospital were taken as the training set, and 1686 patients from Xiangya Second Hospital for the validation set. AKI was developed in 176 (3.5%) and 67 patients (4%) (Table 1), in the training set and the verification set, respectively. The baseline features of low and high NLR group were compared in Table 1. The LASSO regression was used to reduce the number of the variables. Model I was built with the following risk factors: preoperative high NLR, RBC, lymphocyte absolute value, neutrophil absolute value, platelet

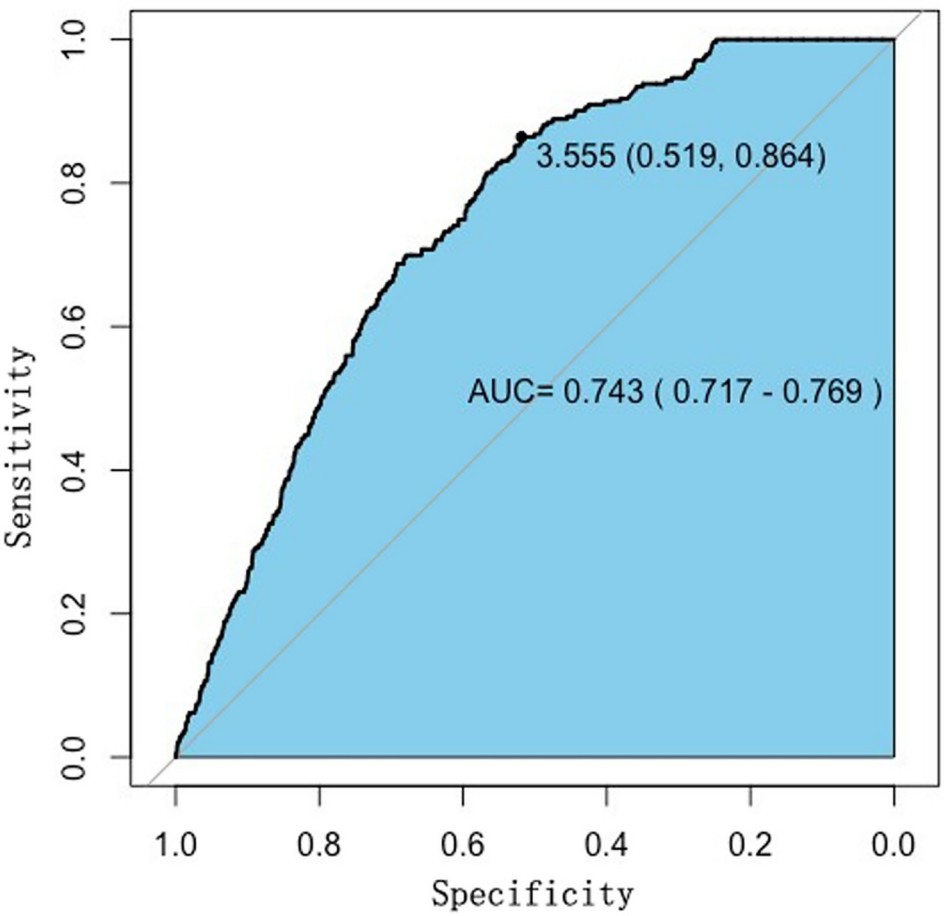

**Fig 1. ROC curve analysis of preoperative Neutrophil/lymphocyte ratio for the occurrence postoperative AKI.**

distribution width, albumin, total bilirubin, estimated 10-year survival rate, emergency, amount of blood loss, and ASA III or IV. The results showed that red blood cells, the absolute value of lymphocytes, albumin, and estimated 10 survival rate played a protective role. While preoperative high NLR was an independent risk factor associated with AKI after non-cardiac surgery (OR, 2.410; 95% CI: 1.371–4.335) (Table 2).

The AUC in the training set was 0.817 (95% CI: 0.784–0.85), while the validation set was 0.804 (95% CI: 0.749–0.85), the AUC deviation was 0.012 (P = 0.68 > 0.05) (Fig 2). The validation showed that preoperative high NLR was still an independent risk factor associated with postoperative AKI.

Moreover, the likelihood ratio test is used to verify the established model. Model 1 was established as above, Model II was built the same as Model I but the risk factor preoperative high NLR was removed. The likelihood ratio test showed Model I was significant different with Model II (P < 0.05). This suggest that preoperative high NLR might be of important value in predicting AKI occurrence after non-cardiac surgery.

## Discussion

In our study, the NLR measured on admission, was associated with the development of postoperative AKI in patients who underwent non-cardiac surgery under general anaesthesia. The

**Table 1. Variables in training set and validation set.**

| Variable | P | Training set Low NLR (N = 2556) | High NLR (N = 2501) | Total (N = 5057) | P | Validation set Low NLR (N = 849) | High NLR (N = 837) | Total (N = 1686) |
|---|---|---|---|---|---|---|---|---|
| **Postoperative AKI, *n*(%)** | 0 | 24(0.9) | 152(6.1) | 176(3.5) | 0 | 9(1.1) | 58(6.9) | 67(4.0) |
| **Male, *n*(%)** | 0 | 1136(44.4) | 1382(55.3) | 2518(49.8) | 0 | 388(45.7) | 462(55.2) | 850(50.4) |
| **Age, M (IQR)** | 0.245 | 51(44,61) | 52(43,62) | 52(43,62) | 0.304 | 51(43,61) | 52(43,62) | 51(43,61) |
| **Hypertension, *n*(%)** | 0.368 | 1062(41.5) | 1007(40.3) | 2069(40.9) | 0.219 | 372(43.8) | 341(40.7) | 713(42.3) |
| **CHD, *n*(%)** | 0 | 135(5.3) | 220(8.8) | 355(7.0) | 0.023 | 42(4.9) | 65(7.8) | 107(6.3) |
| **DM, *n*(%)** | 0.023 | 391(15.3) | 443(17.7) | 834(16.5) | 0.181 | 143(16.8) | 163(19.5) | 306(18.1) |
| **Peripheral vascular disease, *n*(%)** | 0.738 | 241(9.4) | 228(9.1) | 469(9.3) | 0.809 | 81(9.5) | 76(9.1) | 157(9.3) |
| **ASA grade, *n*(%)** | 0 | | | | 0 | | | |
| I, II | | 1904(74.5) | 1394(55.7) | 3298(65.2) | | 641(75.5) | 441(52.7) | 1082(64.2) |
| III | | 614(24.0) | 915(36.6) | 1529(30.2) | | 198(23.3) | 329(39.3) | 527(31.3) |
| IV | | 38(1.5) | 192(7.7) | 230(4.5) | | 10(1.2) | 67(8.0) | 77(4.6) |
| **Estimated 10-year survival rate, M (IQR)** | 0.001 | 0.96(0.90,0.98) | 0.96(0.78,0.98) | 0.96(0.90,0.98) | 0.01 | 0.96(0.90,0.98) | 0.96(0.78,0.98) | 0.96(0.90,0.98) |
| **WBC, M (IQR)** | 0 | 6.0(5.0,7.2) | 10.8(8.0,14.5) | 7.5(5.7,11.0) | 0 | 6.1(5.0,7.3) | 10.7(8.2,14.3) | 7.6(5.7,11.0) |
| **RBC, M (IQR)** | 0 | 4.3(4.0,4.7) | 4.0(3.5,4.5) | 4.2(3.7,4.6) | 0 | 4.3(4.0,4.7) | 4.0(3.5,4.6) | 4.2(3.8,4.7) |
| **Hb, M (IQR)** | 0 | 128(116,139) | 119(101,133) | 124(109,137) | 0 | 128(115,140) | 119(101,136) | 125(108,138) |
| **HCT, M (IQR)** | 0 | 39.2(35.8,42.3) | 36.3(31.0,40.5) | 37.9(33.5,41.6) | 0 | 39.3(35.4,42.6) | 36.1(31.5,41.4) | 37.9(33.4,42.0) |
| **PDW, M (IQR)** | 0 | 14.3(12.4,16.0) | 14.8(12.6,16.2) | 14.6(12.5,16.1) | 0.027 | 14.3(12.5,16.1) | 14.7(12.7,16.2) | 14.5(12.6,16.1) |
| **Neutrophils, M (IQR)** | 0 | 3.5(2.8,4.4) | 9.1(6.4,12.6) | 5.1(3.4,9.2) | 0 | 3.5(2.8,4.5) | 8.9(6.5,12.5) | 5.2(3.5,8.9) |
| **Percentage of neutrophils, M (IQR)** | 0 | 60.1(53.8,65.1) | 84.1(77.1,89.6) | 70.5(59.9,84.0) | 0 | 60.0(54.1,65.6) | 84.2(77.3,90.1) | 70.6(59.8,84.1) |
| **Lymphocyte, M (IQR)** | 0 | 1.8(1.5,2.2) | 1.0(0.7,1.4) | 1.4(1.0,1.9) | 0 | 1.8(1.5,2.2) | 1.0(0.7,1.3) | 1.4(0.9,1.9) |
| **Percentage of lymphocytes, M (IQR)** | 0 | 29.8(24.8,35.3) | 9.4(5.7,14.5) | 20.1(9.5,29.9) | 0 | 29.5(25.0,35.1) | 9.5(5.6,14.8) | 20.3(9.5,29.8) |
| **TBIL, M (IQR)** | 0 | 13.2(10.0,17.6) | 15.0(10.7,21.9) | 13.9(10.3,19.5) | 0 | 13.3(10.0,17.7) | 14.9(11.0,21.6) | 14.2(10.5,19.4) |
| **DBIL, M (IQR)** | 0 | 4.3(3.1,6.0) | 5.6(3.8,8.7) | 4.8(3.4,7.2) | 0 | 4.3(3.1,6.0) | 5.5(3.7,8.6) | 4.9(3.3,7.2) |
| **Albumin, M (IQR)** | 0 | 41.2(38.4,44.2) | 37.5(32.2,42.0) | 39.8(35.6,43.4) | 0 | 41.5(38.5,44.4) | 37.9(32.6,41.9) | 40.0(35.8,43.4) |
| **Globulin, M (IQR)** | 0 | 26.2(23.5,29.1) | 25.0(21.8,28.4) | 25.7(22.7,28.8) | 0 | 26.1(23.4,29.0) | 25.2(22.0,28.6) | 25.7(22.7,28.8) |
| **A/G, M (IQR)** | 0 | 1.6(1.4,1.8) | 1.5(1.3,1.7) | 1.5(1.3,1.8) | 0 | 1.6(1.4,1.8) | 1.5(1.3,1.7) | 1.5(1.3,1.7) |
| **Urea, M (IQR)** | 0.02 | 4.5(3.5,5.5) | 4.4(3.2,5.8) | 4.4(3.4,5.6) | 0.497 | 4.5(3.6,5.6) | 4.5(3.3,5.9) | 4.5(3.5,5.7) |
| **Uric acid, M (IQR)** | 0 | 265(209,324) | 218(154,290) | 245(183,309) | 0 | 265(211,334) | 222(158,295) | 247(183,316) |
| **Creatinine, M (IQR)** | 0.006 | 63(54,74) | 62(51,75) | 63(53,75) | 0.104 | 64(54,76) | 63(52,75) | 64(53,76) |
| **eGFR, M (IQR)** | 0 | 103.4(92.7,112.6) | 105(93.2,115.7) | 104.1(92.9,114.1) | 0.034 | 102.9(91.6,112.6) | 104.9(92.3,114.9) | 103.9(91.8,113.8) |
| **Emergence, *n*(%)** | 0 | 160(6.3) | 786(31.4) | 946(18.7) | 0 | 48(5.7) | 260(31.1) | 308(18.3) |
| **Surgical grading, *n*(%)** | 0.059 | | | | 0.95 | | | |
| 1 | | 1004(39.3) | 902(36.1) | 1906(37.7) | | 317(37.3) | 307(36.7) | 624(37.0) |
| 2 | | 1474(57.7) | 1515(60.6) | 2989(59.1) | | 507(59.7) | 504(60.2) | 1011(60.0) |
| 3 | | 78(3.1) | 84(3.4) | 162(3.2) | | 25(2.9) | 26(3.1) | 51(3.0) |
| **Laparoscope, *n*(%)** | 0 | 810(31.7) | 486(19.4) | 1296(25.6) | 0 | 261(30.7) | 177(21.1) | 438(26.0) |
| **Operative time, M (IQR)** | 0 | 2.6(1.6,3.7) | 2.7(1.8,3.9) | 2.7(1.7,3.8) | 0.08 | 2.7(1.7,3.8) | 2.7(1.9,3.8) | 2.7(1.8,3.8) |
| **Amount of fluid infusion, M (IQR)** | 0 | 25(16,36) | 26(17,36) | 26(16,36) | 0.21 | 26(16,36) | 26(17,36) | 26(16,36) |
| **Amount of blood loss, M (IQR)** | 0 | 2.0(0.5,4.0) | 3.0(1.0,6.0) | 2.0(1.0,5.0) | 0 | 2.0(0.8,4.0) | 2.5(1.0,5.0) | 2.0(1.0,5.0) |

*(Continued)*

**Table 1.** (Continued)

| Variable | P | Low NLR (N = 2556) | High NLR (N = 2501) | Total (N = 5057) | P | Low NLR (N = 849) | High NLR (N = 837) | Total (N = 1686) |
|---|---|---|---|---|---|---|---|---|
| | | Training set | | | | Validation set | | |
| Admission to ICU, *n*(%) | 0 | 58(2.3) | 365(14.6) | 423(8.4) | 0 | 9(1.1) | 144(17.2) | 153(9.1) |

Data are shown as mean ± SD or number (%), as appropriate. P = 0 means P<0.001, a statistically significant difference. AKI, acute kidney injury; Age(years old); CHD, coronary heart disease; DM, diabetes mellitus; ASA, American Society of Anesthesiologists; WBC, White Blood cell($10^9$/L); RBC, red blood cell($10^{12}$/L); Hb, hemoglobin (g/L); Hct, hematocrit(%); PDW, platelet distribution width(fL); Neutrophils($10^9$/L); Lymphocyte($10^9$/L); percentage(%);TBIL, total bilirubin(μmol/L); DBIL, Direct Bilirubin(μmol/L); albumin, globulin(g/L); urea, creatinine, uric acid(μmol/L); eGFR, estimated glomerular filtration rate(ml/min/1.73m²); operative time(hours), intraoperative fluid infusion and blood loss($10^2$ml).

cut-off value of the NLR was 3.55, the sensitivity of identifying AKI was 86.4%, and the specificity was 51.9%. Preoperative high NLR was an independent risk factor associated with AKI after non-cardiac surgery (OR, 2.410; 95% CI: 1.371–4.335).

Inflammation plays an important role in the development of AKI [20]. Some inflammatory marker can predict AKI, such as IL-18 [21], IL-10 [22], IL-6 [22] and α-1 microglobulin [23]. However, these indicators are not carried out as routine examinations in various hospitals; on the other hand, some are only measured postoperatively related to postoperative AKI, which cannot achieve an early warning, so as to carry out intervention in the perioperative period.

The NLR which can be calculated using data from preoperative blood routine test, is a reliable marker for the systemic inflammatory response. Because the number of neutrophils reflects inflammation in the body, and the number of lymphocytes represents the body's response to oxidative stress [24]. In the inflammatory response, lymphocytes can be apoptotic, while neutrophils proliferate. Therefore, to a certain extent, the NLR indicates the balance of the inflammatory and anti-inflammatory reactions. A large number of studies have reported that high NLR is closely related to the development and prognosis of various diseases, such as coronary artery disease, cancer and other diseases [25–27]. In the field of nephorology, previous studies have found that increased preoperative NLR was associated with AKI in patients

**Table 2. The result from the lasso regression.**

| | Risk factor | coefficient | SD | Statistic | P | OR | 2.5%CI | 97.5%CI |
|---|---|---|---|---|---|---|---|---|
| intercept | - | -3.082 | 0.792 | -3.893 | 0 | 0.046 | 0.01 | 0.216 |
| $X_1$ | Preoperative high NLR | 0.88 | 0.293 | 3.005 | 0.003 | 2.41 | 1.371 | 4.335 |
| $X_2$ | RBC | -0.272 | 0.116 | -2.344 | 0.019 | 0.762 | 0.606 | 0.956 |
| $X_3$ | Lymphocyte absolute value | -0.18 | 0.152 | -1.187 | 0.235 | 0.835 | 0.613 | 1.11 |
| $X_4$ | Neutrophil absolute value | 0.031 | 0.016 | 1.928 | 0.054 | 1.031 | 0.999 | 1.063 |
| $X_5$ | Platelet distribution width | 0.083 | 0.027 | 3.03 | 0.002 | 1.086 | 1.028 | 1.145 |
| $X_6$ | Albumin | -0.023 | 0.013 | -1.833 | 0.067 | 0.977 | 0.954 | 1.002 |
| $X_7$ | TBIL | 0.004 | 0.002 | 2.692 | 0.007 | 1.004 | 1.001 | 1.007 |
| $X_8$ | Estimated 10-year survival rate | -1.21 | 0.388 | -3.119 | 0.002 | 0.298 | 0.142 | 0.652 |
| $X_9$ | Emergency | 0.485 | 0.181 | 2.681 | 0.007 | 1.624 | 1.137 | 2.312 |
| $X_{10}$ | Amount of Blood loss | 0.026 | 0.007 | 3.461 | 0.001 | 1.026 | 1.011 | 1.041 |
| $X_{11}$ | ASA III | 0.75 | 0.194 | 3.857 | 0 | 2.116 | 1.45 | 3.113 |
| | ASA IV | 1.166 | 0.272 | 4.294 | 0 | 3.21 | 1.873 | 5.443 |

P = 0 means P<0.001, a statistically significant difference. Abbreviation: OR (odds ratio), CI (confidence interval). RBC (red blood cell) ($10^{12}$/L), absolute values of lymphocytes and neutrophils ($10^9$/L), Platelet distribution width (fL), albumin (g/L), TBIL (total bilirubin) (μmol/L), blood loss ($10^2$ml).

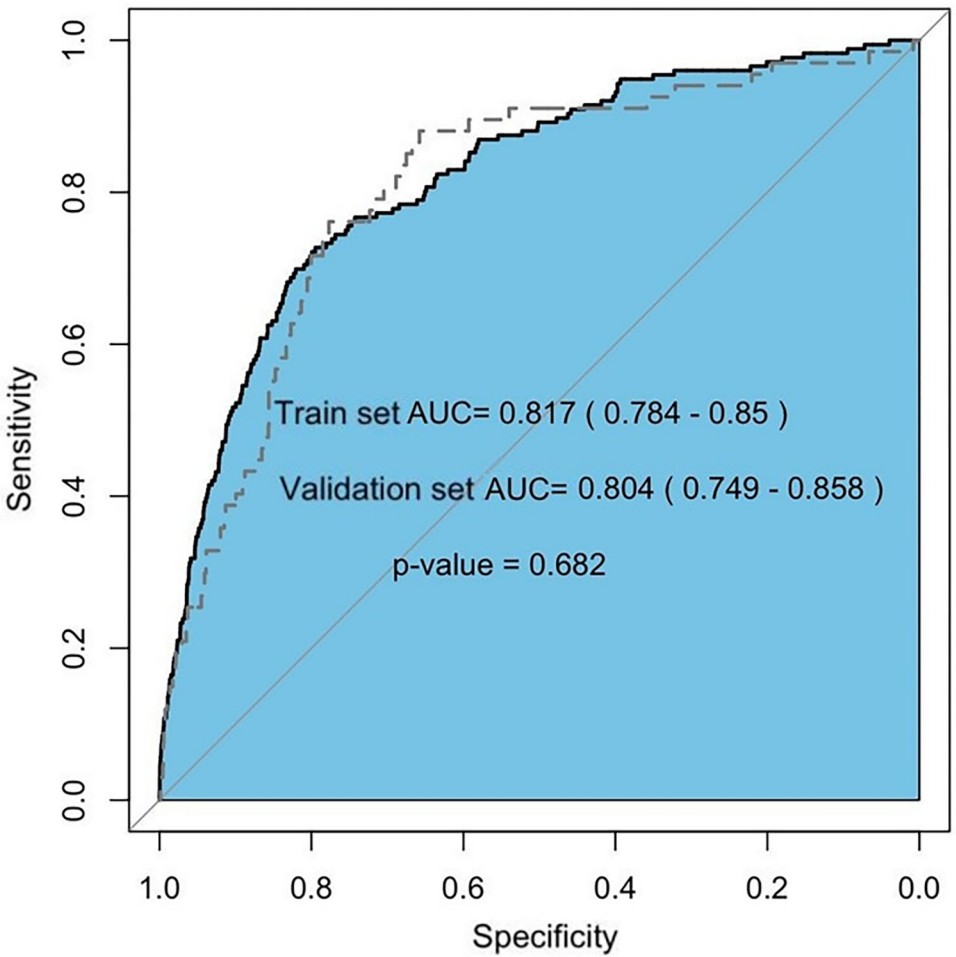

**Fig 2. Relationships between the ROC of training set and validation set.**

who underwent cardiovascular surgery and burn surgery [15, 28, 29], Bu et al. [17] also reported that the initial NLR measured at ICU admission was associated with the development of AKI in patients with sepsis and septic shock. The patients mentioned above are at high risk for postoperative AKI, however, our study included a wider range of patients. Unlike these studies, Yu and his colleagues [30] focused on the postoperative NLR and found that NLR within 24 h after surgery was significantly correlated with major postoperative complications in patients undergoing carotid endarterectomy. By using the optimal cut-off value of preoperative NLR obtained in our study, the rate of missed diagnosis is lower, which enables clinicians to be more alert to the occurrence of postoperative AKI in patients, strengthening preoperative prevention and management (e.g., avoid the use of nephrotoxic drugs or angiography) of postoperative renal function problems.

The association of preoperative high NLR with postoperative AKI may be the result of neutrophil activation prior to surgery. This, may further lead to endothelial injury and activation of coagulation pathway after surgery, which can stimulate the body to produce inflammatory mediators, induce systemic inflammatory response, and lead to postoperative AKI. Our study focused on the effect of preoperative basic state of patients on postoperative AKI, but also included intraoperative patient conditions, such as hemodynamic issues. There were

significant differences in many indicators between the high and low NLR groups, and the factor of P < 0.1 was included to correct these indicators. The method of LASSO regression, which is a compression estimation method based on reducing variable set, is adopted to obtain a concise and effective model. A multi-factor regression model (AUC = 0.817) was constructed after excluding the interference of age, hypertension, surgical type and other factors. Moreover, through the likelihood ratio validation, it was proved again that preoperative high NLR was an independent risk factor for AKI after non-cardiac surgery, and that preoperative high NLR might have important value in predicting the occurrence of AKI after non-cardiac surgery.

## Limitation

Limitations of this study should also be acknowledged. First, we used the discharge diagnosis to identify diseases, which may not cover all types of CKD patients, including those with minor renal impairment defined by a lower creatinine level. Second, some valuable variables that may be prognostic factors were not evaluated in this study, such as surgical information (operative time and urine output) and inflammatory markers other than the NLR (e.g., cytokines). In view of the above limitations, further research is necessary.

## Conclusion

The present study showed that preoperative high NLR (NLR ≥3.555) was an independent risk factor for postoperative AKI in patients for non-cardiac surgery under general anesthesia (OR, 2.410; 95% CI, 1.371–4.335). As a simple and accessible indicator, preoperative NLR is valuable in differentiating high-risk groups for postoperative AKI.

## Supporting information

**S1 Checklist.**
(DOCX)

## Acknowledgments

We thank Dr. Xing Liu for technical consultation.

**Declarations:** Ethics approval and consent to participate: This study was approved by the ethics committee of the third Xiangya hospital of Central South University (2017-S214). Because of the observational nature of the study, informed consent was waived.

## Author Contributions

**Conceptualization:** Bo Li.

**Writing – original draft:** Yongzhong Tang, Linchong Chen, Lin Yang, Wen Ouyang, Dan Li.

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

35047593

