## [Decision Letter · Decision Letter 0]

8 Mar 2022

PONE-D-21-35925Preoperative Neutrophil-Lymphocyte Ratio for Predicting Surgery-Related Acute Kidney Injury in Non-cardiac Surgery Patients under General Anaesthesia: A retrospective cohort studyPLOS ONE

Dear Dr. Li,

Thank you for submitting your manuscript to PLOS ONE. After careful consideration, we feel that it has merit but does not fully meet PLOS ONE’s publication criteria as it currently stands. Therefore, we invite you to submit a revised version of the manuscript that addresses the points raised during the review process.

We look forward to receiving your revised manuscript.

Kind regards,

Giuseppe Filiberto Serraino, M.D., Ph.D.

Academic Editor

PLOS ONE

Journal Requirements:

Reviewers' comments:

Reviewer's Responses to Questions

**Comments to the Author**

1. Is the manuscript technically sound, and do the data support the conclusions?

Reviewer #1: Yes

Reviewer #2: Yes

2. Has the statistical analysis been performed appropriately and rigorously? 

Reviewer #1: Yes

Reviewer #2: Yes

3. Have the authors made all data underlying the findings in their manuscript fully available?

Reviewer #1: Yes

Reviewer #2: Yes

4. Is the manuscript presented in an intelligible fashion and written in standard English?

Reviewer #1: Yes

Reviewer #2: Yes

5. Review Comments to the Author

Reviewer #1: Thanks to the authors for their efforts and for touching on an important issue in the practice of nephrology.

Although it is a retrospective study, it consolidates the increasing evidence of the utility of NLR in predicting AKI.

I have the following points:

1. Can the authors elaborate more on the pre-op creatinine values?

2. What do you think is the reason for the low specificity of 51.9%?

3. It would be better to mention data about the utility in Cardiovascular procedures, like CABG or repair of AA or in the HBP surgeries.

4. Authors should mention data, if any, about the use of NLR in the post-op setting to compare with their results.

4. It would be better to refer to the following paper, Chinese; with Meta-Analysis data:

Lu Z, Wang L, Jia L, Wei F, Jiang A. [A Meta-analysis of the predictive effect of neutrophil-lymphocyte ratio on acute kidney injury]. Zhonghua Wei Zhong Bing Ji Jiu Yi Xue. 2021 Mar;33(3):311-317. Chinese. DOI: 10.3760/CMA.j.cn121430-20201215-00755. PMID: 33834972.

5. The meta-analysis mentioned above has provided data about ethnic differences in NLR values (Asia vs. Eurasia). The authors can compare their cohort with the other groups

Reviewer #2: The neutrophil to lymphocyte ratio in various diseases had been discussed in detail in the past years. However, this important work is the first to suggest that it's preoperative value may predict postoperative AKI in non-cardiac surgery.

There are two papers that may increase the value of "Discussion" if mentioned.

The work of Yu Y, et al. described similar results, but the NLR was measured postoperatively in the first 24 hours. They studied a much smaller number of patients.

Yu Y, Cui WH, Cheng C, Lu Y, Zhang Q, Han RQ. Association between neutrophil-to-lymphocyte ratio and major postoperative complications after carotid endarterectomy: A retrospective cohort study. World J Clin Cases. 2021 Dec 16;9(35):10816-10827. doi: 10.12998/wjcc.v9.i35.10816. PMID: 35047593; PMCID: PMC8678856.

The paper of Guangqing Z, et al. found the same result but in cardiac surgery. This recent paper probably was not published before the preparation of this paper.

Guangqing Z, Liwei C, Fei L, Jianshe Z, Guang Z, Yan Z, Jianjun C, Ming T, Hao C, Wei L. Predictive value of neutrophil to lymphocyte ratio on acute kidney injury after on-pump coronary artery bypass: a retrospective, single-center study. Gen Thorac Cardiovasc Surg. 2022 Feb 1. doi: 10.1007/s11748-022-01772-z. Epub ahead of print. PMID: 35103920.

A few spelling mistakes:

line 94: Diabetes;

line 136: Low;

line 142: Preoperative do not need capital letter.

6. PLOS authors have the option to publish the peer review history of their article (what does this mean?). If published, this will include your full peer review and any attached files.

Reviewer #1: No

Reviewer #2: No

---

## [Author Response · Author response to Decision Letter 0]

11 Apr 2022

Re. MS # PONE-D-21-35925

Dear Giuseppe Filiberto Serraino:

We are submitting a revision of our manuscript entitled “Preoperative Neutrophil-Lymphocyte Ratio for Predicting Surgery-Related Acute Kidney Injury in Non-cardiac Surgery Patients under General Anaesthesia: A retrospective cohort study” for possible publication in PLOS ONE. We thank the Editor and Reviewers for their thoughtful and constructive comments on the initial submission. In response to the comments, we have addressed their critiques point-by-point as detailed below.

Reviewer #1: Thanks to the authors for their efforts and for touching on an important issue in the practice of nephrology.

Although it is a retrospective study, it consolidates the increasing evidence of the utility of NLR in predicting AKI.

A: We thank the Reviewer for the comments and have addressed the Reviewer’s other concerns below.

Q1: Can the authors elaborate more on the pre-op creatinine values? 

A: The baseline serum creatinine level was calculated using the lowest level at preoperative day 7 and detected by the L-type creatinine M (Fujifilm Wako Pure Chemical Corporation, Japan).

Q2. What do you think is the reason for the low specificity of 51.9%? 

A: The cut-off value of NLR differs in different patients (PMID: 33834972), the specificity and sensitivity also varies. The possible reason of the low specificity in our study might be the cut-off value and the type of surgery. 

Q3. It would be better to mention data about the utility in Cardiovascular procedures, like CABG or repair of AA or in the HBP surgeries. 

A: Thanks for your suggestion! We added the results of the NLR in cardiovascular surgery in our manuscript (Page 10, Main Document - marked copy).

Q4. Authors should mention data, if any, about the use of NLR in the post-op setting to compare with their results. 

A: Thanks for your advice! We mentioned the study of postoperative NLR in the revised manuscript (Page 10, Main Document - marked copy).

Q5. It would be better to refer to the following paper, Chinese; with Meta-Analysis data: Lu Z, Wang L, Jia L, Wei F, Jiang A. [A Meta-analysis of the predictive effect of neutrophil-lymphocyte ratio on acute kidney injury]. Zhonghua Wei Zhong Bing Ji Jiu Yi Xue. 2021 Mar;33(3):311-317. Chinese. DOI: 10.3760/CMA.j.cn121430-20201215-00755. PMID: 33834972.

A: Thanks for your suggestion! We added this reference in our manuscript (Page 10, Main Document - marked copy).

Q6. The meta-analysis mentioned above has provided data about ethnic differences in NLR values (Asia vs. Eurasia). The authors can compare their cohort with the other groups.

A: The meta-analysis showed that increased NLR had predictive value for the occurrence of AKI in patients from Southeast Asia (MD = 4.04, 95%CI was 1.09-6.99, P = 0.007) and Eurasia (MD = 2.51, 95%CI was 1.12-3.90, P = 0.0004). The patients in our study were all from China, and the results were similar with the meta-analysis.

Reviewer #2: The neutrophil to lymphocyte ratio in various diseases had been discussed in detail in the past years. However, this important work is the first to suggest that it's preoperative value may predict postoperative AKI in non-cardiac surgery.

A: We thank the Reviewer for the comments and have addressed the Reviewer’s other concerns below.

There are two papers that may increase the value of "Discussion" if mentioned.

The work of Yu Y, et al. described similar results, but the NLR was measured postoperatively in the first 24 hours. They studied a much smaller number of patients.

Yu Y, Cui WH, Cheng C, Lu Y, Zhang Q, Han RQ. Association between neutrophil-to-lymphocyte ratio and major postoperative complications after carotid endarterectomy: A retrospective cohort study. World J Clin Cases. 2021 Dec 16;9(35):10816-10827. doi: 10.12998/wjcc.v9.i35.10816. PMID: 35047593; PMCID: PMC8678856.

The paper of Guangqing Z, et al. found the same result but in cardiac surgery. This recent paper probably was not published before the preparation of this paper.

Guangqing Z, Liwei C, Fei L, Jianshe Z, Guang Z, Yan Z, Jianjun C, Ming T, Hao C, Wei L. Predictive value of neutrophil to lymphocyte ratio on acute kidney injury after on-pump coronary artery bypass: a retrospective, single-center study. Gen Thorac Cardiovasc Surg. 2022 Feb 1. doi: 10.1007/s11748-022-01772-z. Epub ahead of print. PMID: 35103920.

A: Thanks for your suggestion! We added these two papers in our Discussion section (Page 10, Main Document - marked copy).

A few spelling mistakes:

line 94: Diabetes;

line 136: Low;

line 142: Preoperative do not need capital letter.

A: We are sorry about the mistake! We corrected these spelling mistakes in the revised version (Page 6, 8, Main Document - marked copy)

Once again, we thank the Editor and Reviewers for critically reviewing this study and for your constructive comments. We also make minor changes to our funding information in our revised manuscript. We look forward to hearing from you in the near future.

Sincerely,

Dan Li

Department of Anesthesiology

The Third Xiangya Hospital, Central South University

138 Tongzipo Road, Changsha, 

Hunan，410013, China

---

## [Decision Letter · Decision Letter 1]

3 Jun 2022

Preoperative Neutrophil-Lymphocyte Ratio for Predicting Surgery-Related Acute Kidney Injury in Non-cardiac Surgery Patients under General Anaesthesia: A retrospective cohort study

PONE-D-21-35925R1

Dear Dr. Li,

We’re pleased to inform you that your manuscript has been judged scientifically suitable for publication and will be formally accepted for publication once it meets all outstanding technical requirements.

Kind regards,

Giuseppe Filiberto Serraino, M.D., Ph.D.

Academic Editor

PLOS ONE

Additional Editor Comments (optional):

Reviewers' comments:

Reviewer's Responses to Questions

**Comments to the Author**

1. If the authors have adequately addressed your comments raised in a previous round of review and you feel that this manuscript is now acceptable for publication, you may indicate that here to bypass the “Comments to the Author” section, enter your conflict of interest statement in the “Confidential to Editor” section, and submit your "Accept" recommendation.

Reviewer #2: All comments have been addressed

2. Is the manuscript technically sound, and do the data support the conclusions?

Reviewer #2: Yes

3. Has the statistical analysis been performed appropriately and rigorously? 

Reviewer #2: Yes

4. Have the authors made all data underlying the findings in their manuscript fully available?

Reviewer #2: Yes

5. Is the manuscript presented in an intelligible fashion and written in standard English?

Reviewer #2: Yes

6. Review Comments to the Author

Reviewer #2: Good revision. All comments have been addressed. The suggested literature was incorporeted and the spelling mistakes corrected.

7. PLOS authors have the option to publish the peer review history of their article (what does this mean?). If published, this will include your full peer review and any attached files.

Reviewer #2: No

---

## [Editor Report · Acceptance letter]

21 Jul 2022

PONE-D-21-35925R1 

Preoperative Neutrophil-Lymphocyte Ratio for Predicting Surgery-Related Acute Kidney Injury in Non-cardiac Surgery Patients under General Anaesthesia: A retrospective cohort study 

Dear Dr. Li:

I'm pleased to inform you that your manuscript has been deemed suitable for publication in PLOS ONE. Congratulations! Your manuscript is now with our production department. 

Kind regards, 

on behalf of

Professor Giuseppe Filiberto Serraino 

Academic Editor

PLOS ONE